# Expressibility of OWL Axioms with Patterns

Aaron Eberhart✉, Cogan Shimizu, Sulogna Chowdhury, Md. Kamruzzaman
Sarker, and Pascal Hitzler

Data Semantics Lab, Kansas State University, USA
**aaroneberhart@ksu.edu**

**Abstract.** The high expressivity of the Web Ontology Language (OWL)
makes it possible to describe complex relationships between classes, roles,
and individuals in an ontology. However, this high expressivity can be an
obstacle to correct usage and wide adoption. Past attempts to amelio-
rate this have included the development of specific, presumably human-
friendly syntaxes, such as the Manchester syntax or graphical interfaces
for OWL axioms, albeit with limited success. If modelers want to develop
suitable OWL axioms it is important to make this as easy as possible.
In this paper, we adopt an idea from the Protégé plug-in, OWLAx, which
provides a simple, clickable interface to automatically input axioms of a
limited number of types by following simple axiom patterns. In particu-
lar, each of these axiom patterns contains at most three classes or roles.
We hypothesize that most of the axioms in existing ontologies could
be expressed semantically in terms of simple patterns like these, which
would mean that more complex patterns can be used very sparingly.
Our findings, based on an analysis of 518 ontologies from six public
ontology repositories, confirm this hypothesis: Over 90% of class axioms
in the average ontology are indeed expressible with our simple patterns.
We provide a detailed analysis of our findings.

## 1 Introduction

Knowledge graph schema are complex artifacts that can be difficult and ex-
pensive to produce and maintain. This is especially true when encoding them
in OWL (the Web Ontology Language) as ontologies. The high expressivity of
OWL is a boon, in that it makes it possible to describe complex relationships
between classes, roles,[1] and individuals in an ontology. At the same time, how-
ever, this high expressivity is often an obstacle to its correct usage that can
limit adoption. Past attempts to ameliorate this have included the development
of specific, presumably human-friendly syntaxes, such as the Manchester syntax
[10], or graphical interfaces for OWL axioms, albeit with modest success [16].
Additionally, certain engineering paradigms and methodologies have been devel-
oped, such as eXtreme Design [2] or Modular Ontology Modeling [7,19], that try
to simplify the modeling process.

---

[1] We refer to properties as **roles**, unless a distinction is relevant, as this is the standard
description logic term. These include both object properties and data properties.

In general, these methodologies aim to guide ontology developers through the complex modeling process by either abstracting the complexity away (for example, through the use of Ontology Design Patterns), or by limiting the scope of the model to something immediately applicable and understandable. In this paper we are particularly interested in the latter, especially during the axiomatization process. We believe that it is important to investigate new avenues for improving the approachability of creating suitable OWL axioms.

One of the core tenets of the Modular Ontology Modeling methodology is to produce schema diagrams and then systematically axiomatize them, with the input of domain experts. This systematic axiomatization is inspired by the OWLAx plugin for Protégé,[2] which provides a simple, clickable interface to automatically input axioms of limited syntactic forms that are all created from simple axiom patterns [17]. In particular, each of these simple axiom patterns contains at most three classes or roles. In [17], it was posited (but not demonstrated) that the 17 axiom patterns provided by the interface were sufficient for *most* modeling purposes. In this paper, we test that hypothesis by analyzing 518 ontologies from six public ontology repositories. Concretely, we show the following:

**H1.** Almost all axioms in OWL ontologies are expressible using a set of simple axiom patterns, like those found in Table 1.

And indeed, as we will see, it holds for over 90% of class axioms in the average ontology using our relatively straightforward analysis. With a more thorough analysis or with different patterns, the percentage may even be higher.

## 2   Related Work

We are aware of only a very limited amount of research that specifically concerns the semantic, not syntactic, composition and expressibility of ontologies regarding patterns. There are several studies, such as [5,23,14], which investigate the use of OWL syntax and constructs in general. However, a mere syntactic survey of OWL as it is used in practice does not directly address the question we are investigating, namely whether a relatively small set of axiom patterns suffices to express most OWL axioms. Zhang et al. [26] look at ways to measure the design complexity of ontologies. Their work is focused more on ontology quality evaluation than ontology composition. Some have also attempted to measure the effect that axioms like existential quantifiers have on reasoning time, such as Kang et al. [11], although it is only tangentially related to the work that we are presenting.

There are also, as previously mentioned, tools that attempt to simplify OWL ontology development, such as Manchester Syntax [10], WebVOWL [13], CoModIDE [18], Graffoo [3], and ROWLTab [16]. These tools simplify the development process but they do not measure whether OWL axioms are necessarily complex

---

[2] See https://protege.stanford.edu/.

in everyday usage. It could very well be the case that OWL is unavoidably complicated and these tools are needed to deal with this complexity, although we believe our work demonstrates that this is usually not the case.

## 3    Methodology

Our hypothesis is that most axioms in ontologies could be expressed with simple axiom patterns. In this section, we will define what we mean by simple axioms, then give an example of a set of patterns that generate simple axioms, such as those used in the Protégé plugin OWLAx. Following that, we will describe how to measure the extent to which an ontology is expressible with simple axiom patterns, and then provide some minimal normalizations for ontologies which we will use in our evaluation.

### 3.1    Simple Axioms

The simple axioms we study in this paper are defined below. We consider description logic syntax for OWL DL, that is, we identify it with the description logic $\mathcal{SROIQ}(D)$ [8].

**Definition 1.** *A **Simple Axiom** is any OWL axiom that contains at most three class or role names, or a data range, and is not a syntactic shortcut for other OWL axioms as defined in the OWL 2 Specification.*[3] *Any axiom which is not simple is a **Complex Axiom**.*

Our set of axiom patterns is designed for class axioms, so we restrict our focus to class axioms in the evaluation, although, in principle, the notion of a simple axiom could apply to role (RBox) axioms as well. The limitation of three atoms for simple axioms is an intuitive threshold, in terms of size, because it means that nesting is limited, yet the axiom can still contain expressions and participate in complex inferences in combination with other simple axioms. This would not be the case for axioms limited to size two, where one could only express $A \sqsubseteq B$ for classes, or $R \sqsubseteq S$ for roles, which would radically limit the expressivity of the ontology. Axioms with more than three atomic classes or roles may be more expressive, but are often equivalent through normalization to smaller axioms, so they do not make not good candidates for simple axioms. Note that negation and inverse are not considered complex, since the definition considers only the number of names; of course double negation can be eliminated trivially.

### 3.2    OWLAx axiom patterns

OWLAx [17] is a Protégé plugin that allows users to automatically generate certain simple OWL axioms using a graphical interface. The set of axioms we study in this paper are inspired by the axioms that OWLAx can create, and they are listed in Table 1.

---

[3] See `https://www.w3.org/TR/2012/REC-owl2-syntax-20121211/`

| | | | |
|---|---|---|---|
| Subclass | $A \sqsubseteq B$ | Functional | $\top \sqsubseteq \leqslant 1R.\top$ |
| Disjoint Classes | $A \sqcap B \sqsubseteq \bot$ | Qualified Functional | $\top \sqsubseteq \leqslant 1R.B$ |
| Domain | $\exists R.\top \sqsubseteq B$ | Scoped Functional | $A \sqsubseteq \leqslant 1R.\top$ |
| Scoped Domain | $\exists R.A \sqsubseteq B$ | Qualified Scoped Functional | $A \sqsubseteq \leqslant 1R.B$ |
| Range | $\top \sqsubseteq \forall R.B$ | Inverse Functional | $\top \sqsubseteq \leqslant 1R^-.\top$ |
| Scoped Range | $A \sqsubseteq \forall R.B$ | Inverse Qualified Functional | $\top \sqsubseteq \leqslant 1R^-.B$ |
| Existential | $A \sqsubseteq \exists R.B$ | Inverse Scoped Functional | $A \sqsubseteq \leqslant 1R^-.\top$ |
| Inverse Existential | $A \sqsubseteq \exists R^-.B$ | Inverse Qualified Scoped Functional | $A \sqsubseteq \leqslant 1R^-.B$ |
| | | Structural Tautology | $A \sqsubseteq \geqslant 0R.B$ |

A,B,R are variable terms that contain at most one class or role name, or a data range

Table 1: OWLAx Axiom Patterns

The actual implementation details of the OWLAx plugin are not pertinent to our discussion. Rather, we are interested in what it happens to contain: a set of patterns that only make simple axioms. In this sense, the axiom patterns are *simple axiom patterns*, since they can generate only simple axioms. And because it was designed specifically to help create ontologies, we speculate that ontologies will be mostly expressible using these patterns. We now discuss how to assess the extent to which an ontology can be expressed using such *axiom patterns*.

### 3.3   Axiom Pattern Expressibility

To study whether axioms in an ontology are expressible with simple axiom patterns, we first define the term axiom pattern and then show how a set of axiom patterns can be used to study axioms in an ontology, obtaining multiple metrics to evaluate pattern expressibility.

**Definition 2.** *An **Axiom Pattern** is a programmatic template for creating new, syntactically correct axioms. An axiom pattern may have variable terms that can be used to obtain specific axioms by substitution. Given an axiom $\alpha$ and a pattern p, we say that p can generate $\alpha$ (or, p is $\alpha$-generating) if $\alpha$ can be obtained by appropriately substituting variable terms in p.*

For example, the axiom pattern $A \sqsubseteq \exists R.B$ for an existential axiom from Table 1, where A, B, R are variable terms, can be used to generate the axiom Dog $\sqsubseteq$ $\exists$chases.Squirrel by substitution, where "Dog" and "Squirrel" are classes and "chases" is a role. Note that axiom patterns are very different from Ontology Design Patterns (ODPs) [20], because an ODP is a partial ontology representing a generic solution to a recurring ontology modeling problem, while an axiom pattern is a pattern for making single axioms. They are fundamentally different in purpose and nature (although the term *pattern* can be used for either).

**Definition 3.** *The **Axiom Pattern Expressibility** $ae_{\mathcal{P}}(\alpha)$ of an axiom $\alpha$ w.r.t. a set of axiom patterns $\mathcal{P}$ is the set of patterns $p \in \mathcal{P}$ each of which can generate $\alpha$ with the fewest substitutions. Formally, given an axiom $\alpha$ and a*

pattern $p$ that can generate $\alpha$, let $s_p(\alpha)$ be the number of substitutions required to generate $\alpha$ from $p$. Given an axiom $\alpha$, and a set $\mathcal{P}$ of axiom patterns, let $ae_{\mathcal{P}}(\alpha)$ be the set of $\alpha$-generating patterns from $\mathcal{P}$ such that, for all $q \in ae_{\mathcal{P}}(\alpha)$ and $p \in \mathcal{P}$, we have $s_q(\alpha) \leq s_p(\alpha)$. Note that $ae_{\mathcal{P}}(\alpha)$ may be empty if there are no $\alpha$-generating patterns in $\mathcal{P}$.

We give an example for these definitions. Let $P$ be the set of axiom patterns from Table 1, and let $\alpha$ be Human $\sqsubseteq \, \leq 1.\text{hasHeart}.\top$. Then $s_{A \sqsubseteq \leq 1R.\top}(\alpha) = 2$ and $s_{A \sqsubseteq \leq 1R.B}(\alpha) = 3$. It is easy to check that no other patterns in $P$ can generate $\alpha$. Hence $ae_{\mathcal{P}}(\alpha) = \{A \sqsubseteq \, \leq 1R.\top\}$.

**Proposition 1.** *For $\mathcal{P}$ the set of axiom patterns from Table 1, and given any axiom $\alpha$ in $\mathcal{SROIQ}(D)$, $ae_{\mathcal{P}}(\alpha)$ is either empty or a singleton set. I.e. if an axiom can be generated from a pattern from $P$, then there is a unique pattern with the minimal number of required substitutions.*

*Proof.* It is sufficient to show that no axiom can be created by more than one of our patterns with the fewest substitutions. This can be verified easily by inspecting Table 1 and comparing pairs of patterns. We draft the thought process. Table 1 states that each variable term contains at most one class or role name, therefore subclass, disjoint classes, and structural tautology patterns all have no overlapping patterns which could also produce axioms of those forms. The domain, range, and existential patterns do not mutually overlap except in pairs that vary only in the location of $\top$ or $^-$, so the claim is clearly also true, since $\top$ or $^-$ in the pattern reduces the number of substitutions required to generate an axiom containing it by 1. Functional and inverse functional patterns follow a similar structure, where an axiom is always uniquely obtainable with the fewest substitutions from the pattern containing $\top$ and $^-$ in the same locations.

The following will be used in our evaluations.

**Definition 4.** *The **Average Axiom Pattern Expressibility** $\overline{ae}_{\mathcal{P}}(\mathcal{A})$ for a set of axioms $\mathcal{A}$ of cardinality $|\mathcal{A}|$ (i.e., $|\mathcal{A}|$ is the number of axioms in $\mathcal{A}$) is defined as*

$$\overline{ae}_{\mathcal{P}}(\mathcal{A}) = \frac{1}{|\mathcal{A}|} \sum_{\alpha \in \mathcal{A}} |ae_{\mathcal{P}}(\alpha)|.$$

*The **Average Ontology Axiom Pattern Expressibility** $\overline{oe}_{\mathcal{P}}(\mathcal{O})$ of a set of ontologies $\mathcal{O}$ having cardinality $|\mathcal{O}|$ and set of axiom patterns $\mathcal{P}$, is given by*

$$\overline{oe}_{\mathcal{P}}(\mathcal{O}) = \frac{1}{|\mathcal{O}|} \sum_{\mathcal{A} \in \mathcal{O}} \overline{ae}_{\mathcal{P}}(\mathcal{A}),$$

*where $\mathcal{A}$ represents the set of axioms in each ontology.*

It is important to note that average ontology axiom pattern expressibility can be used to evaluate a set of ontologies, and average axiom pattern expressibility can be used to evaluate any number of axioms, e.g. a set of axioms which has been collected from multiple ontologies.

### 3.4   Normalization

We have discussed our evaluation measures for axiom pattern expressibility of an ontology. However, there remains an issue that ontologies often vary radically in the way they are syntactically expressed, even if semantically they mean similar or even equivalent things. Ontologies are written for completely different purposes and at differing levels of complexity; some ontologies are developed for complex reasoning applications, while others are used for more straightforward data integration. Even within a single ontology, different authors may express equivalent statements in different ways based on personal preference or style. To give an example, class disjointness of two classes $A$ and $B$ can be expressed with any of the (equivalent) axioms $A \sqcap B \sqsubseteq \bot$, $A \sqsubseteq \neg B$, $B \sqsubseteq \neg A$, and others.

Hence, in order to evaluate the pattern expressibility of a large number of ontologies uniformly, we therefore need at least a minimal syntactic normalization strategy taken from community standards that allows us to compare disparate sources without biasing the evaluation in favor of any particular style. For this, we use multiple strategies derived from common OWL practices.

Our normalization begins by filtering out all axioms except class, role, and HasKey axioms. This is necessary because there are many OWL axioms for which our pattern study will not apply. Included in this are assertion (ABox) axioms, since these are primarily axioms about instances rather than classes and roles, but also axioms such as annotations, declarations, and datatype definitions, that are axioms according to the OWL 2 specification but carry no or few formal semantics. HasKey axioms are taken into account because they are logical axioms and not assertions or annotations, although their semantics is different from class and role axioms. Note that none of our axiom patterns matches HasKey axioms. The remaining class and role axioms are then transformed according to the following procedures.

The first transformation that we perform is an equivalence transformation based on the syntactic shortcuts defined in the OWL Structural Specification [15]. Whenever an axiom is found that has one of the forms in Column 1 of Table 2, we perform the designated substitution. These substitutions are equivalent rewritings so they do not alter the semantics of the ontology. It is also possible that other simple transformations of class axioms according to the equivalences defined in the structural specification could improve the evaluation, since these axioms also might be expressible using patterns. Our transformation thus may lead to an undercount in our disfavor; we will come back to this point later. Thus, for EquivalentClasses, DisjointClasses, and DisjointUnion we convert them to sets of SubClass axioms using definitions in the OWL 2 specification.

The second transformation that we perform is obtaining negation normal form (NNF) of all class axioms in an ontology. By using the NNF we can transform all of the class axioms in an ontology into simple syntactic forms that are stripped of unnecessary information that might be due to coincidence rather than semantic equivalence.

The last transformation we apply is splitting SubClass axioms with conjunctions in the consequent, or disjunctions in the antecedent, into separate axioms.

| Ontology Axiom | Substituted Axiom |
|---|---|
| ReflexiveObjectProperty(R) | $\top \sqsubseteq \exists R.\mathbf{Self}$ |
| IrreflexiveObjectProperty(R) | $\exists R.\mathbf{Self} \sqsubseteq \bot$ |
| FunctionalObjectProperty(R) | $\top \sqsubseteq\ \leqslant 1R.\top$ |
| FunctionalDataProperty(S) | $\top \sqsubseteq\ \leqslant 1S.\top$ |
| InverseFunctionalObjectProperty(R) | $\top \sqsubseteq\ \leqslant 1R^-.\top$ |
| ObjectPropertyRange(R C) | $\top \sqsubseteq \forall R.C$ |
| DataPropertyRange(S D) | $\top \sqsubseteq \forall S.D$ |
| ObjectPropertyDomain(R C) | $\exists R.\top \sqsubseteq C$ |
| DataPropertyDomain(S C) | $\exists S.\top \sqsubseteq C$ |

R is an ObjectProperty, S is a DataProperty, C is a Class, and D is a DataRange

Table 2: Axiom Transformations

This is a standard procedure in many normalizations, and we simply replace the axiom with a set of axioms formed from the conjuncts or disjuncts whenever an axiom of this type is found. There is a special case that occurs only when the consequent is an ExactCardinality expression whose value is equal to 1. In this case, we do not use a MinCardinality 1 substitution but instead add an existential, since that is equivalent and more compact.

All normalizations are performed sequentially and axioms are output into a separate collection for evaluation. This compartmentalizes the data and ensures that no duplicates are created, even when a single axiom is transformed into a set of axioms. The sets of axioms $\mathcal{A}$ used in our evaluations are these separate normalized sets.

## 4 Evaluation

We analyze a set of 518 ontologies from various sources, normalizing them, and testing them for axiom pattern expressibility according to the principles described in the previous section. Ontologies were selected from diverse sources with unique design requirements: benchmark ontologies, Ontology Design Patterns (ODPs), ontologies extracted from Linked Open Vocabulary (LOV) [22], as well as medical domain ontologies. It would technically be possible to integrate other types of linked data in this analysis, though semantically it may not be straightforward to interpret the results if the data was mixed, so we use only OWL ontologies. Statistics about the original ontologies gathered before and after normalization can be found in Table 3. The normalization adds a few axioms to the set of axioms from an ontology whenever an axiom is split, so counts increase by a factor of around 7-8 to 10, except for ontologies that contain many assertions that were excluded during the normalization process, like hydrography and anatomy benchmarks. As mentioned previously, the normalized axioms are the axioms used in the evaluation and the equations. We use the set of all normalized axioms from all ontologies for the $\overline{ae}_{\mathcal{P}}(\mathcal{A})$ numbers and the set of normalized axioms from each ontology for $\overline{oe}_{\mathcal{P}}(\mathcal{O})$ numbers.

|                   | LOV    | Hydrography | Anatomy | Conference | ODP  | Ontobee | Misc   |
|-------------------|--------|-------------|---------|------------|------|---------|--------|
| classes           | 20075  | 341         | 6048    | 498        | 577  | 509925  | 80796  |
| roles             | 10106  | 121         | 5       | 226        | 600  | 10453   | 535    |
| data properties   | 8340   | 77          | 0       | 85         | 71   | 480     | 66     |
| axioms            | 274991 | 8873        | 41407   | 3037       | 7378 | 5071892 | 816572 |
| logical axioms    | 82450  | 5463        | 16383   | 2153       | 3223 | 963880  | 272334 |
| normalized axioms | 94937  | 1748        | 9951    | 2774       | 3917 | 1208950 | 391015 |
| ontologies        | 250    | 4           | 2       | 7          | 80   | 171     | 4      |

Table 3: Ontology Statistics. Logical axioms are axioms that are neither declarations nor annotations

In this section, we report the result for all ontologies we tested, then go into details about each source, reporting a separate evaluation for each. Next we break down the results by profile and report the numbers for those as well. In all cases, the expressibility numbers are reported for All Axioms, Class Axioms, and Simple Class Axioms. Our axiom patterns can only express simple class axioms, thus the values for 'All Axioms' represent the evaluation using all class, role, and HasKey axioms, 'Class Axioms' indicates the evaluation using only class axioms, and 'Simple Class Axioms' represents the evaluation for only simple class axioms. In Figures and Tables, the term "miss" is used to indicate a simple axiom that was observed but was inexpressible using our patterns. The last value we report, which is a byproduct of calculations that produce expressibility numbers, is the percent subclass and percent existential, as well as their combination. By this we mean, what percent of all of the axioms in an ontology are expressible with the subclass pattern, the existential pattern, or both. It will turn out in nearly every case that a surprisingly high proportion of most ontologies is expressible with just these two simple axiom patterns. OWL files and source code for the evaluation, except the gene ontology which can't be uploaded due to size restrictions, can be found on the GitHub page `https://github.com/aaronEberhart/owlax` and the raw data can be inspected in the spreadsheet at `https://tinyurl.com/eswc2021`.

### 4.1   Overall Expressibility

The average axiom pattern expressibility and the average ontology axiom pattern expressibility for our simple axiom patterns over all normalized axioms in all ontologies is included in Table 4, as well as the standard deviation for the ontology axiom pattern expressibility.

Figure 1 shows the overall distribution of axioms for the entire collection of ontologies. Complex class axioms, role axioms, miss (inexpressible simple axioms) and HasKey are the axioms that cannot be generated by our patterns; note that only the first two of these play a significant role. Simple subclass is 54.8%, and existential is 23.9%, totaling 78.8%. This is almost the same as the axiom expressibility value for all axioms (82.2%). A more detailed view of axiom type distributions can be found in the next section in Figure 2.

| | $\overline{ae}_{\mathcal{P}}(\mathcal{A})$ | $\overline{oe}_{\mathcal{P}}(\mathcal{O})$ | StdDev $\overline{oe}_{\mathcal{P}}(\mathcal{O})$ |
|---|---|---|---|
| All Axioms | 82.2% | 82.9% | 0.206 |
| Class Axioms | 83.8% | 92.9% | 0.165 |
| Simple Class Axioms | 99.8% | 96.5% | 0.153 |

Table 4: Overall Average Expressibility

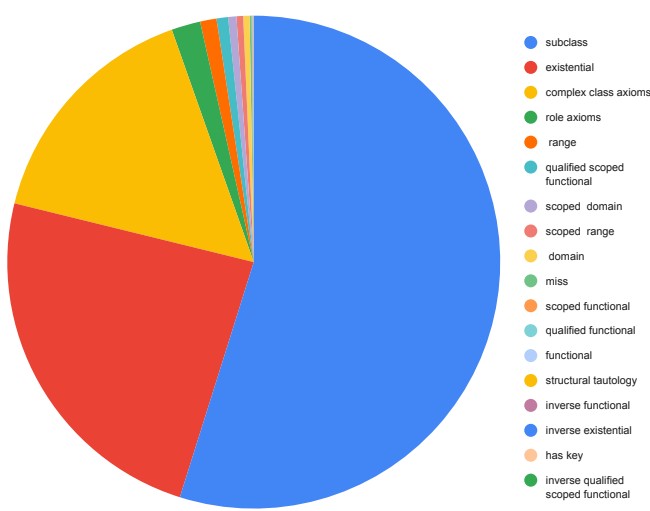

Fig. 1: Overall Distribution of Axioms

## 4.2   Source Expressibility

As they are all from very different domains, each source was analyzed independently from the whole. Our evaluation includes 250 ontologies that were automatically pulled from LOV using a script that can be found on the project GitHub. We obtained benchmark ontologies that are used for ontology alignment evaluation. There are 4 ontologies from Hydrography, 2 from Anatomy, and 7 from the Conference domains, and each appear in their own column in Tables 5 and 6. We also obtained and evaluated 80 ODPs, as well as a collection of 171 OWL files that are mainly from the medical domain from the ontobee[4] [25] website. Additionally, we gathered 4 ontologies that did not fall neatly into any of these categories but nonetheless seemed interesting to include in the overall result. These ontologies are General Formal Ontology [6], Gene Ontology [4], GeoLink Base Ontology [12], and the Enslaved Ontology [21], and their average is labeled Misc in the tables.

---

[4] http://ontobee.org.

|                     | LOV   | Hydrography | Anatomy | Conference | ODP   | Ontobee | Misc  |
|---------------------|-------|-------------|---------|------------|-------|---------|-------|
| All Axioms          | 78.7% | 77.1%       | 99.9%   | 89.3%      | 85.7% | 88.7%   | 62.4% |
| Class Axioms        | 92.6% | 84.5%       | 100%    | 96.2%      | 97.3% | 89.9%   | 62.5% |
| Simple Class Axioms | 99.2% | 96.5%       | 100%    | 99.2%      | 99.1% | 99.8%   | 99.9% |

Table 5: $\overline{ae}_{\mathcal{P}}(\mathcal{A})$ By Source

|                     | LOV   | Hydrography | Anatomy | Conference | ODP   | Ontobee | Misc  |
|---------------------|-------|-------------|---------|------------|-------|---------|-------|
| All Axioms          | 81.3% | 77.5%       | 99.9%   | 87.3%      | 76.5% | 88.3%   | 67.2% |
| Class Axioms        | 92.7% | 89.7%       | 100%    | 92.8%      | 96.4% | 91.8%   | 99.4% |
| Simple Class Axioms | 95.2% | 97.6%       | 100%    | 95.2%      | 97.7% | 97.9%   | 99.4% |

Table 6: $\overline{oe}_{\mathcal{P}}(\mathcal{O})$ By Source

The Gene Ontology tends to dominate the other sources in Misc due to its extremely large size. It also contains a much higher percentage of complex class axioms than any other ontology we tested, which accounts for the difference in Misc between simple class axioms and class axioms. In LOV there are a considerable number of role axioms. This explains why the expressibility is so much higher for class axioms than all axioms.

In Table 7, we see the range of percent subclass and existential among sources. Anatomy, Ontobee, and Misc all contain medical domain ontologies, which may account for the increase in percent existential if they contain more ontologies in the EL profile. LOV and Hydrography, on the other hand, are expressible with very little subclass at all, and both contain many role axioms. Except for the Anatomy Benchmarks, which is actually only two ontologies so a disproportionately small sample size, it does not appear to be the case that any sources are entirely existential and subclass. Neither are any sources completely lacking the two axiom patterns. When we break the results down by profile in the next section, things will look quite a bit different.

Figure 2 shows the actual counts of each axiom pattern used to calculate expressibility in logarithmic scale. In this chart we can see how sources like Ontobee and Misc do contain some of the less common patterns. They are just so large that smaller sources, like ODPs and benchmarks, tend to have higher percentages. The previously mentioned high percentage of complex class axioms for Misc can be seen in the third column. Also, the two ontology sources with the highest percent expressibility of subclass and existential are Anatomy and Ontobee, both medical type ontology sources. If we move farther down the chart to the less common axiom patterns, the larger ontology sources are less prevalent and now the benchmarks and ODPs start to dominate. The last two axiom patterns were never detected by our program and inverse scoped functional occurred once so the log scale in the chart hides this. For the inverse qualified functional axiom pattern it is conceivable that authors rarely had occasion to write axioms like this. Disjoint classes may seem surprising, however we investigated the evaluation and found that, even though our pattern, $A \sqcap B \sqsubseteq \bot$,

| | LOV | Hydrography | Anatomy | Conference | ODP | Ontobee | Misc |
|---|---|---|---|---|---|---|---|
| Subclass | 42.4% | 46.2% | 66.8% | 57.6% | 56.4% | 60.2% | 41.0% |
| Existential | 05.9% | 07.4% | 33.1% | 06.7% | 05.7% | 26.5% | 20.7% |
| Subclass + Existential | 48.3% | 53.3% | 99.9% | 64.3% | 62.1% | 86.7% | 61.7% |

Table 7: Percent Subclass and Existential By Source

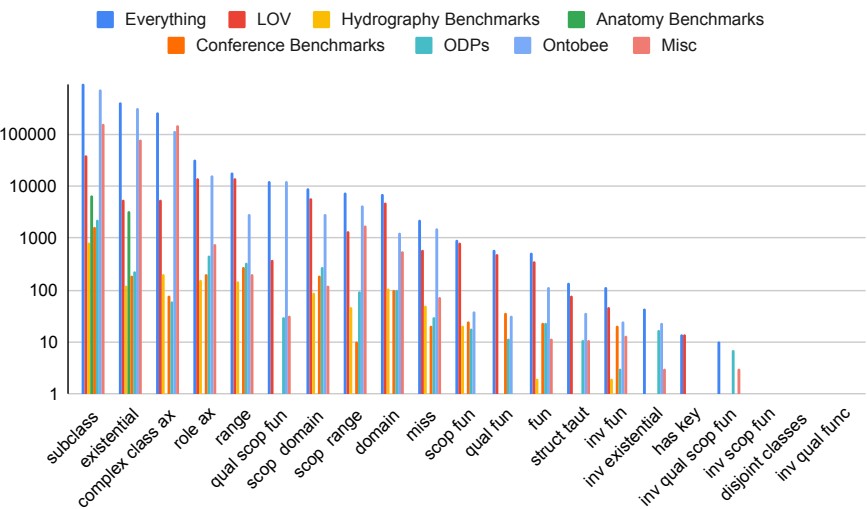

Fig. 2: Axiom Expressibility Counts, Log Scale

is expressible in profiles that do not contain negation, authors are likely using Protégé or the OWLAPI [9] to state disjoint classes axioms which the normalization transforms into subclass axioms containing negation, as defined in the specification. This causes disjoint classes to match the subclass pattern rather than our disjointness pattern, which is not a false negative or a methodological error, but it is technically a misclassification due to conflicting sets of patterns.

### 4.3   Profile Expressibility

During the analysis we also tested each ontology to see if it was in the OWL profiles EL, QL, RL, or DL, and report the expressibility information for each profile separately. In Tables 8, 9 and 10, the Full column is reproduced from values in Section 4.1 and Table 4 for comparison, since every ontology will be in OWL Full. There were 15 ontologies that could be loaded into the evaluation, but the OWLAPI could not test their profile; these ontologies were not included in the profile results but are included in the Full column. Interestingly, for the EL, QL, and RL profiles we see around a ten percent expressibility boost over the overall result. All three also have perfect expressibility for simple class axioms,

|                      | EL     | QL     | RL     | DL     | Full   |
|----------------------|--------|--------|--------|--------|--------|
| All Axioms           | 98.7%  | 99.7%  | 97.8%  | 91.4%  | 82.2%  |
| Class Axioms         | 98.8%  | 100%   | 100%   | 92.7%  | 83.8%  |
| Simple Class Axioms  | 100%   | 100%   | 100%   | 99.9%  | 99.8%  |

Table 8: $\overline{ae}_{\mathcal{P}}(\mathcal{A})$ By Profile

|                      | EL     | QL     | RL     | DL     | Full   |
|----------------------|--------|--------|--------|--------|--------|
| All Axioms           | 94.6%  | 87.7%  | 81.7%  | 82.8%  | 82.9%  |
| Class Axioms         | 97.5%  | 97.1%  | 95.2%  | 94.4%  | 92.9%  |
| Simple Class Axioms  | 98.1%  | 97.1%  | 95.2%  | 97.2%  | 96.5%  |

Table 9: $\overline{oe}_{\mathcal{P}}(\mathcal{O})$ By Profile

and nearly perfect expressibility for all class axioms. The expressibility numbers for DL are also slightly higher than the overall numbers, though significantly less so than for the other profiles. The lower values for Full compared to DL are heavily influenced by the Gene Ontology, which was not classified as OWL DL.

Unlike the different sources, where the percent subclass and existential numbers were mostly near the average, we get a much more skewed result when we break the ontologies down by profile in Table 10. EL and DL ontologies seem to be expressible with a similar percentage of subclass axioms as the overall result, though EL has many more existential expressions. QL ontologies, on the other hand, are eighty percent expressible with the simple subclass pattern. And the RL profile ontologies are almost entirely expressible with simple subclass. It is no surprise, then, that EL, QL, and RL ontologies have such high expressibility.

In Table 11, we mark which of our axiom patterns are expressible in each profile with an X symbol, using the OWL 2 Profiles [24] document as a reference. We see that RL expressibility is almost entirely subclass, and the existential pattern is indeed inexpressible in that profile. EL seems to be evenly divided between subclass and existential, which again aligns with the types of statements permitted in the profile. The DL profile allows all the types of expressions and it understandably has a similar result to the overall average.

For the EL profile we also observe a unique result, because our axiom patterns have almost complete overlap with the 4 normal form class axioms defined for $\mathcal{EL}^{++}$ in [1], as shown in Table 12. The only exception is conjunction, which can only match our disjoint classes axiom pattern when the consequent is equal to $\perp$. If we were to define a conjunction axiom pattern, it might be possible to completely express this profile with simple axiom patterns. This could also be done for class axioms in the QL profile, where our axiom patterns could express all simple axioms, and could express any set of QL axioms that was normalized to remove nested quantifiers. Simple class axioms for the RL profile could be expressed in much the same way as EL, missing only conjunction axioms that do not have $\perp$ in the consequent. With the addition of a conjunction pattern

|  | EL | QL | RL | DL | Full |
|---|---|---|---|---|---|
| Subclass | 52.5% | 78.2% | 95.1% | 60.9% | 54.8% |
| Existential | 46.1% | 21.1% | 0% | 29.2% | 23.9% |
| Subclass + Existential | 98.6% | 99.3% | 95.1% | 90.2% | 78.8% |

Table 10: Percent Subclass and Existential By Profile

| | EL | QL | RL | DL | | EL | QL | RL | DL |
|---|---|---|---|---|---|---|---|---|---|
| $A \sqsubseteq B$ | X | X | X | X | $\top \sqsubseteq\, \leqslant 1R.\top$ | | | | X |
| $A \sqcap B \sqsubseteq\, \bot$ | X | | X | X | $\top \sqsubseteq\, \leqslant 1R.B$ | | | | X |
| $\exists R.\top \sqsubseteq B$ | X | X | X | X | $A \sqsubseteq\, \leqslant 1R.\top$ | | | X | X |
| $\exists R.A \sqsubseteq B$ | X | | X | X | $A \sqsubseteq\, \leqslant 1R.B$ | | | X | X |
| $\top \sqsubseteq\, \forall R.B$ | | | | X | $\top \sqsubseteq\, \leqslant 1R^-.\top$ | | | | X |
| $A \sqsubseteq\, \forall R.B$ | | | X | X | $\top \sqsubseteq\, \leqslant 1R^-.B$ | | | | X |
| $A \sqsubseteq\, \exists R.B$ | X | X | | X | $A \sqsubseteq\, \leqslant 1R^-.\top$ | | | X | X |
| $A \sqsubseteq\, \exists R^-.B$ | | X | | X | $A \sqsubseteq\, \leqslant 1R^-.B$ | | | X | X |
| | | | | | $A \sqsubseteq\, \geqslant 0R.B$ | | | | X |

Table 11: Profile Axiom Pattern Expressibility

and by normalizing to remove nested quantifiers we could also obtain complete class axiom pattern expressibility for RL.

## 5  Discussion

Our motivation for this study is that we believe *simple axioms*, specifically those that can be created from simple patterns, are easier for non-logicians to understand and utilize for modeling. This is not to say that complex axioms are unnecessary, indeed they may be the most important. However, if most of OWL could be expressed with simple patterns, as we have shown, this seems a good place to focus our attention when we consider ways to facilitate adoption. Alongside improved comprehension, simple axioms made from patterns come with a number of added benefits: attempting to measure the non-local effects of ontological commitments may be easier, they can be easily and automatically created by tools that allow users to specify statements in a graphical interface without a deep technical understanding of the inner-workings of OWL, and simple axioms often do not require normalization before being input to a reasoner. To support this, we determine the current usage characteristics of axioms in existing ontologies to see if they are expressible with simple axiom patterns, as well as how this relates to different sources and OWL profiles.

Our evaluation is limited in a few ways. We have not yet investigated if limiting an ontology engineer to these axiom patterns would pose additional obstacles, such as in writing complex axioms. There is also no way we are aware of to automatically detect if patterns were used to make an ontology, so we are unable to compare ontologies made without patterns to those that were made with patterns. Finally, our evaluation produces a lower bound. So while it shows

| Axiom Pattern | $\mathcal{EL}^{++}$ Normal Class Axiom |
|---|---|
| A ⊑ B | A ⊑ B |
| A ⊓ B ⊑ ⊥ | A ⊓ B ⊑ C, when C = ⊥ |
| ∃R.⊤ ⊑ B | ∃R.A ⊑ B, when A = ⊤ |
| ∃R.A ⊑ B | ∃R.A ⊑ B |
| A ⊑ ∃R.B | A ⊑ ∃R.B |

Table 12: $\mathcal{EL}^{++}$ Axiom Pattern Expressibility

clearly that axiom patterns can provide sufficient expressibility for most axioms, we do not yet know how much more a more sophisticated evaluation might find.

### 5.1  Future Work

In the future there are many potential next steps that could build on this study. One approach would be to test different sets of simple axiom patterns and see how the expressibility numbers compare between them. OWLAx was a good basis to create an initial set of simple axiom patterns but there are some obvious common ones that it lacks, for instance conjunction, disjunction, negation, as well as multiple variations on cardinality and role axioms. For the current study we only use simple axioms because there is no clearly defined way to categorize complex axioms, which can be arbitrarily large. It may be interesting to analyze the complex axioms to see if there are any new patterns that can be included.

We also admit that our definition of expressibility is quite simple, intentionally kept this way for clarity. However it may be possible with some more comprehensive statistical tools that a better understanding of axiom pattern expressibility in ontologies is possible. In a future study we may look into different evaluations besides expressibility, perhaps it will be informative to compare.

Additionally, our method normalized many axioms, however it is likely that complex axioms existed in the ontologies we studied that *could* have been normalized but *weren't* because our method only obtained NNF and then split up appropriate conjunction and disjunction axioms. By introducing new terms in the normalization to syntactically split some expressions we might be able to even further increase the expressibility detection capability. Though, as previously mentioned, this would require the addition of new terms, and would be equivalent but also contain more entities, so the comparison would be less obviously appropriate.

## 6  Conclusion

In this paper we demonstrate that most axioms in OWL ontologies are expressible with a small set of simple axiom patterns. This has implications for how we approach ontology management and development. If ontologies are mostly expressible with simple patterns then focusing on supporting and explaining these types of axiom patterns can lead to easier adoption and maintenance. Complex

axioms will of course always be a part of OWL, but we can improve our ontologies most easily by first making sure that the patterns used to create simple axioms are well understood and used correctly.

*Acknowledgement* The authors acknowledge partial support from the financial assistance award 70NANB19H094 from U.S. Department of Commerce, National Institute of Standards and Technology and partial support from the National Science Foundation under Grant No. 2033521.

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
