# OpenReview forum: "Expressibility of OWL Axioms with Patterns"
_eswc-conferences.org/ESWC/2021/Conference/Research_Track — ESWC 2021 Research_

### Official Review · AnonReviewer1 · 2021-01-11
**A significant result but presentation should be improved**

**Rating:** 1
**Confidence:** 4
**Impact:** 4
**Design And Technical Quality:** 3

**Review:**

This paper demonstrates empirically that a short of axioms (simple axioms summarized in Table 1) is much more frequent in real ontologies: 84% of the axioms, covering 93% of the ontologies. The evaluation is restricted to class axioms. The approach seems technically correct, is original, and significant: the results could have consequences in the design of ontology tools such as ontology editors, OWL 2 justifications, etc.

To make the paper self-contained, the main families of OWL 2 axioms, and the definition of class axioms should be recalled to the reader. Furthermore, it seems to me that the authors use "role" to denote an "object property", but roles are often used to denote properties (abstract role for object properties and concrete roles for data properties). In page 4 the authors also use "S" to denote an object property, I suggest doing the same thing all over the paper: one symbol for properties and another one for object properties.

There is some confusion between "simple axiom" and "simple axiom pattern in Table 1". The evaluation mentions simple axioms but actually Table 1 axioms are meant, I strongly suggest using a specific name for the axioms in Table 1.

The summary that the result "holds for over 90% of class axioms" should be clarified, to avoid the confusion between axiom expressability and ontology expressability. As it is written, the sentence seems to mean axiom expressability, but the sentence is only true for ontology expressability according to Table 4.

At first, it seemed surprising to me that class assertion axioms were not evaluated. The reason that "the notion of an axiom pattern has little relevance for a fact" did not help. I would say that in practice all class assertions are of the form a : A, with A being a concept name, so they are simple. However, by including this easy case, the expressability of "all axioms" would notably increase.



*********************************************

Update: Thank you very much for the rebuttal.


**Anonymity:**

Yes, I would like my review to remain anonymous.

**Reuse And Availability:**

1: Very low

**Strong Points:**

- The demonstration that simple axioms are much more frequent could have consequences in ontology tools.

- Evaluation considers a number of ontologies from different datasets and seems significant.


**Subreviewer:**

I submitted this review.

**Weak Points:**

- Presentation and terminology (class axiom, role, simple axiom) can be confusing.

---

> ### Author Rebuttal · Authors · 2021-01-29
>
> Thank you for your comments, we enjoy thinking about some of your suggestions. We will respond to a few specific remarks:
>
> "it seems to me that the authors use "role" to denote an "object property""\
> We understand that role is synonymous with property and concept synonymous with class as is typical in description logic and OWL. We mean to say role instead of property in the paper generally, unless specifically discussing situations where the presence of data properties may be relevant, so as to avoid confusion and improve formatting of small figures by avoiding the longer words. We will state this explicitly and correct instances where the synonyms may be switched unintentionally.
>
> "I would say that in practice all class assertions … are simple."\
> This is an interesting point! We could also internalize class assertions and study them this way. However, an assertion would be something like a pattern for instances, rather than a pattern for classes or roles, so it would need some thought how to accommodate that change… Also, if facts are already typically expressed with simple patterns, do we really need to evaluate how else they could have been expressed in the first place?
>
> "There is some confusion between "simple axiom" and "simple axiom pattern in Table 1"."\
> In our definitions we distinguish between an axiom, which is a complete entity, and an axiom pattern, which may be used to create an axiom, but is not necessarily itself a complete axiom. We will clarify that a pattern may also be simple. Table 1 contains the axiom patterns derived from OWLAx, all of which are named and can only produce simple axioms as indicated in the last sentence of 3.2.

---

### Official Review · AnonReviewer2 · 2021-01-12
**Interesting Study but in Need of Improvements in Formalization**

**Rating:** 1
**Confidence:** 4
**Impact:** 3
**Design And Technical Quality:** 3

**Review:**

The paper addresses the interesting question of what kind of expressivity ontologies actually use by evaluation ontologies from various sources.

Throughout the paper, the authors work with the premise that simpler axioms are better because non-logicians (speak domain experts) can understand them. But that makes the implicit assumption that a domain ontology can always be developed by domain experts without the help of ontologists (with logic experience) - which goes counter to the thinking in at least part of the ontology engineering community which accepts that good ontologies only emerge from the collaboration between domain experts and ontologists.
Rather, might the restriction of many existing ontologies not be the result of tool limitations? Precisely because certain tools only support certain types of axioms (e.g. the ones that can be easily constructed by click and select -- without really typing anything -- in Protege) we find mostly such kind of axioms? That doesn't prove that more expressive axioms shouldn't be included.

I would also like to see the method underlying the work to be implemented in a reusable tool so that others can run it over their own ontology or a set of ontologies or to reproduce the results. This shouldn't be difficult (I would expect the authors have everyone in this regards implemented) but it is not mentioned at all in the paper.

Some questions and comments:
p 2, Sec 2, first sentence: this seems to imply that the presented work is a investigating the expressibility of ontologies semantically and not just syntactically. I believe this is technically wrong, because only certain kind of syntactic variations are addressed

p 3, Def 1: "and is not a syntactic shortcut for other OWL axioms"
- this needs to be phrased more clearly
- after the definition: "nesting in the expression is limited to at most one quantifier" - why is that? I don't see that. Do you mean "one existential quantifier"? Even that would not be obvious and requires some justification

p. 4: Def 2 needs to be made much more precise to be of any value

p 4/5, starting after Def. 3: You make the assumption that the axiom pattern expressibility is at most 1. This is far from obvious and needs a proof, especially because the whole evaluation depends on this fact.
But if you can show that this 0 or 1, it might be wise to define the subsequent definitions treating it as a Boolean rather than a set (with assumed maximum cardinality of 1).

p 5: last sentence of Sec. 3.3: "an axiom can only ever match at most one of our axiom patterns" -- this is not obvious either and needs to proved (see above)

p 6, 2nd paragraph: need to properly define what class, property, and HasKey axioms are.
In the same paragraph, you mention "annotation axioms, declaration axioms" - I'm confused by this language because logically speaking, these are not axioms at all but "annotations", "declarations", etc.
In the same vein, I don't understand the difference between axioms and logical axioms in Table 3.  For example, we typically distinguish definitions from other axioms, but annotations and declarations are not axioms at all.
In the subsequent Tables, when you refer to "All Axioms", what do you include? I would get rid of annotations and declarations and the-like first because they have nothing to do with the expressivity of the axioms.

p 8, 2-3rd lines: as before, I'd suggest to get rid of the "All Axioms" (unless I misunderstand something, but then please clarify) and you please define "Class Axioms" properly. For the "Simple Axiom" numbers, do you mean "Simple Class Axioms" (as a subset of the Class Axioms) or does it include Property Axioms as well (which would be confusing as not really comparable to the previous numbers)?

p 11: "The program is working correctly ... " - this is an odd statement that should go without saying
the next sentence: "There is just no pattern in our set that can discriminate ... " -- that sounds to me like a contradiction to the earlier statement (p. 5, see my comment above) that each axiom can only match one pattern. If that's not the case, please explain.

p 13, Sec 5: "Out motivation is that we belive simple axioms are easlier for non-logician to understand and utilize for modeling." We can agree on this, but the sole criteria for evaluating an ontology shouldn't be whether non-logicians can understand it. The use of more expressive axioms should be governed by whether they are needed - which you can't base on the fact that 90% of all axioms in some ontologies are simple. What about the other 10%? They might be the most crucial axioms.  The real question is whether and when more expressive axioms are really needed rather than being just a complicated way of expressing multiple simple axioms. This distinction should be made more clear here.
Also, it should be noted that many of the chosen ontologies are lightweight ontologies (as, e.g., LOV and anatomy/medical ontologies are mostly taxonomies with domain and range restrictions) as confirmed withe Fig 2. The ontologies in the MISC category seem to be more representative of ontologies that are more densely axiomatized and actually make use of a larger percentage of the OWL-Full profile.

----
In response to the author's rebuttal:
----
I appreciate the authors taking the time to carefully consider all the comments and questions. Overall, I like the direction of the work but still see a lot of need for more a cleaner and more precise use of language and a discussion of the limitations.  That being said, I think the necessary changes can be feasibly made by the authors (and I would sincerely hope the authors take the time to do that) and thus believe the paper could be accepted.

I just would like to reiterate that if the paper is accepted, the authors need to discuss the limitations in more detail (especially the caveat of limiting designers to certain kind of pattern, which may give them the mistaken perception that these kind of axioms are the only ones one needs to think about.
I'm not quite sure whether my concern about the paper being mislabel as studying the "semantic expressibility" (also raised by another reviewer) has come across correctly. A study of semantic expressibility would need to account for all syntactic variants to express the same content. This is certainly not the cause here because only SOME (i.e. a subset of all possible) syntactic variations are considered. So yes, the transformation preserves the semantic, but some axioms with the same semantics are still being ignored.  This requires careful editing of the wording in some places.

Also, the paper must clarify the terminology so that the reader doesn't need to go digging elsewhere.
Don't assume that every reader is familiar with all the terminology used in the structural specification of OWl or names of classes/methods in OWLAPI.
For example, it would be easy enough to summarize the meaning of a "class axiom" as being one of following kind of statement (you can even refer to the patterns you list):
ClassAxiom := SubClassOf | EquivalentClasses | DisjointClasses | DisjointUnion
Likewise, when first using the term "axioms" and "logical axioms" make clear that annotations, declarations and HasKey statements fall under the former even though they have no "logical content".
But my point is that ClassAxiom, ObjectPropertyAxiom, DataPropertyAxiom should be the ONLY ones considered for the analysis. Everything else is just distracting as it plays no role it determining what percentage of axioms are expressible using simple patterns. Worse, including any nonlogical axioms in the calculations of your percentages would distort the results and make it more difficult to interpret. I could not figure out whether nonlogical axioms are currently included or not.

The terminology in the analysis needs to be consistent as well: does "all axioms" mean "all logical axioms"? I'm not sure.
Also, defined Simple Class Axioms as Class Axioms \ Complex Class Axioms is circular - you technically haven't defined Complex Class Axioms (complex is defined in terms of simple earlier on ...).

About the definition of simple axiom, my point was that the part "Simple Axiom is any OWL axiom that contains at most
three class, role, or data property names," is not sufficiently clear and may not actually say what you want to say.
Let me illustrate: you say that $\forall R. \exists S. C\sqsubset D$ is not simple. But you can still have nested expressions that are simple, e.g. $\forall R. \exists S. D\sqsubset D$ (even if this particular example doesn't make a whole lot of sense, it is after all a valid expression).
Likewise, about your point of "nesting in the expression is limited to at most one quantifier". A subproperty axiom, if I'm not mistaken, technically uses two universal quantifiers but I don't think you want to exclude that! That's why I suggested expressing it differently.

**Anonymity:**

Yes, I would like my review to remain anonymous.

**Reuse And Availability:**

2: Low

**Strong Points:**

- addresses an important question about the kind of expressivity that ontologies actually use, which can inform tool development, reasoning, etc.
- evaluation: evaluates a decent set of ontologies and shows that a large percentage of axioms fit into a small set of simple axiom patterns.


**Subreviewer:**

I submitted this review.

**Weak Points:**

- the presentation is technically sloppy and vague in key places and not sufficiently succinct in its terminology
- because some key terms (class axioms, simple axioms) are not (properly) defined, it is difficult to interpret the percentages in the evaluation; since these are the key contribution, they need to be made more clear
- some assumptions (including about what expressivity is needed by an ontology) need more discussion as they are not necessarily universally accepted

While addressing these weakness requires more theoretical/formalization work with substantial changes to the paper, I expect that they could be addressed in the final version.

---

> ### Author Rebuttal · Authors · 2021-01-29
>
> Thank you for your detailed response to our paper. We agree with many of your comments and think that they can improve our evaluation. We respond to a few:
>
> "and is not a syntactic shortcut..."\
> We will add a footnote here. This is a reference to a term used frequently in the OWL 2 structural specification to denote equivalences that express larger axiom(s) in a more concise way.
>
> "Def 2 needs..."\
> We will add some specificity. It was deliberately left general so as to not restrict it to specific software or technology, but it can be better.
>
> "believe this is technically wrong..."\
> Indeed we perform a semantics-agnostic transformation to syntactically alter the ontology. This allows us to see which of the axioms, even if they are expressed as complex axioms, might have been expressed with simple axioms. The idea, as we say, is that simple axiom(s) can usually express the same semantics in a way that is syntactically easier for developers and domain experts to understand while meaning the same thing.\
> Please also note that more sophisticated pattern sets would only yield a higher expressibility. In this sense, our count is an undercount which is sufficient to show that axiom patterns carry a long way in terms of ability to express content in OWL ontologies.
>
> "need to properly define what class, property, and HasKey axioms..."\
> We will add a footnote here. The terms Class, Property, and HasKey are taken directly from the OWL 2 structural specification, so it is redundant to reproduce that. \
> In OWL 2, Annotations and Declarations are indeed axioms according to the grammar, and the OWLAPI also identifies them as such. However they are not logical axioms, like you point out, which is why we ignore them. The term “logical axiom” is a label repeated from a function in the OWLAPI that means an axiom which is not an annotation or declaration.
>
> "nesting in the expression is limited to..."\
> For example, a nested quantifier would be $\forall R.\exists S.C \sqsubseteq D$ where the quantified expression itself contains a class expression with a quantifier. If we have a limit of 3 class/role names for simple axioms it is impossible to have recursive nesting like this. The nesting refers to the changing scope of the terms in the expression that occurs during quantification, which cannot be achieved in DL with flat conjunction and disjunction alone.
>
> "That doesn't prove that more expressive axioms..."\
> "The use of more expressive axioms ... the most crucial axioms. "\
> We completely agree! We of course don’t mean to imply that the complex axioms aren’t important, they may be the most important. However, if we can syntactically simplify a large portion of the complex axioms without altering their semantics, then we believe that indicates simple axiom patterns may make development easier. The language in the intro and conclusion will be adjusted to make this more clear.
>
> "I would also like to see ... a reusable tool..."\
> Thanks, excellent idea. The software used in the evaluation is on GitHub and linked in the paper. Future work may include development of this into a software resource.
>
> "I'd suggest to get rid of the "All Axioms" ..."\
> The explanatory sentence for these terms may be unclear; we will reword it. The "Class Axioms" value means All Axioms $\setminus$ \{ property axioms $\cup$ has key axioms \} and corresponds to the usual notion, while “Simple Class Axioms” means Class Axioms $\setminus$ Complex Class Axioms.
>
> "You make the assumption that the axiom pattern expressibility is at most 1."\
> This is not an assumption, but we agree this should be demonstrated. We will supply a short proof in the final version. Two of our axiom patterns have been discussed in the paper, the claim can be verified for any OWLAx pattern in the same way. One can check if an arbitrary simple axiom that is expressible with an OWLAx pattern _with the fewest substitutions_ is also expressible with a different OWLAx pattern _with the same number of substitutions_. This is not the case.\
> While it is true that a boolean would simplify the definition, we use sets and their sizes so that it is clear how the different types of patterns are identified within the sum. This also allows greater flexibility in possible future work that may have non-binary expressibility.
>
> "this is an odd statement that should go without saying the next sentence"\
> We will phrase this better. We mean to say that, even though it appears to be a mistake, this is in fact a quirk of the OWLAx pattern set conflicting with typically used patterns. There is no negation in our patterns, so disjointness $A \sqsubseteq \neg B$ can only be expressed with the simple subclass pattern $A \sqsubseteq B$. Adding a $A \sqsubseteq \neg B$ or $\neg A \sqsubseteq B$ pattern in future work would allow the program to discriminate between disjointness with negation and simple subclass. This misclassification is due to the different pattern sets, not an empirical or methodological error.

---

### Official Review · AnonReviewer3 · 2021-01-13
**Review for Semantic Expressibility of OWL Ontologies**

**Rating:** 1
**Confidence:** 4
**Impact:** 3
**Design And Technical Quality:** 3

**Review:**

In this paper a quantitative analysis on the prevalence of "simple" axioms in a number of existing ontologies is presented. While this is interesting, I see a few shortcomings.

The main motivation given is testing the hypothesis that ontology engineers require mostly "simple" axioms.

However, it is unclear to me whether the ontologies examined are actually hand-crafted. Some of them might have been (semi-)automatically generated? But this is a minor issue.

To me a more serious issue in this respect is the normalization step. This essentially means that the examined ontology is no longer in the format in which it was created (in whatever  way).

Another related issue that is partially discussed is the way in which ontology editors might have already influenced the way in which the ontology was actually created.

Then, the authors also mention that the measures are fairly simple and that more sophisticated statistical methods could be used for the analysis.

One could also imagine a qualitative analysis on what kind of statements one cannot express using a given set of axioms - I suppose quite a bit is known already, though.

Overall, this is some interesting work with not overly surprising results and some shortcomings.

Some minor observations:

p.1 schema are -- schemas?

p.2 approachability -- really the best word in this context?

p.2 patterns[15] -- space

p.2 left unproven -- maybe better unproved; but on a semantic level, can this really be proved?

p.2 Almost all -- perhaps be more precise?

p.2 With a more thorough analysis or with different patterns, the percentage may be even higher. -- It is unclear why a more thorough analysis could raise this percentage. Also, I don't think that this kind of speculation should be in the introduction.

p.3 are often equivalent -- analysis when not?

p.3 they do not make not good candidates -- something is wrong here...

p.3 make simple axioms -- create? but it also follows later in the sentence

p.4 Table 1 -- of course one wonders whether there would be more simple axioms than these

p.5 and so on -- could be deleted I think

p.5 n ontologies -- I would not introduce n and just use $| \cal O |$ for better readability

p.5 The last paragraph in 3.3 is repetitive, this was discussed at sufficient length earlier already

p.6 It is also possible that other simple transformations ... could reduce false negative matches -- not really clear

p.7 Table 3 has figures on the ontologies before normalization -- why? And how do these figures look like after normalization?

p.8 percent -- percentage

p.11 The program is working correctly and the reported results are correct. -- bad style; also which "program"?

p.14 consist primarily property axioms -- of



**Anonymity:**

Yes, I would like my review to remain anonymous.

**Reuse And Availability:**

3: Medium

**Strong Points:**

Quantitative study on prevalence of simple axioms in existing ontologies


**Subreviewer:**

I submitted this review.

**Weak Points:**

The normalization step seems to go against the main motivation.

The title suggests a qualitative, not quantitative study.

Evaluation is fairly simple and straightforward (but was certainly tedious).

Significance of the findings is unclear.

---

> ### Author Rebuttal · Authors · 2021-01-29
>
> Thank you for your comments! We appreciate all of the editing tips and will take them into consideration. To address some specific things you mention:
>
> "However, it is unclear to me whether the ontologies examined are actually hand-crafted."\
> This is a good point, we will look into the method of creation for our sources. This may be difficult to determine for some of the automatically extracted sources, but it would definitely be interesting to refine the study this way as well.
>
> "This essentially means that the examined ontology is no longer in the format..."\
> "The normalization step seems to go against the main motivation."\
> We agree that an additional analysis of the non-normalized ontologies could show the potential benefit for simple axioms in a future study, by comparing that result with results from syntactic normalizations. It was presumed that comparing the obviously worse simple pattern expressibility of ontologies as they are with the expressibility of syntactically normalized ontologies would be too trivial a result, but perhaps this is not the case. For the current work we will add information to the tables about the effect of the syntactic transformations on the number of axioms to help illustrate the changes.\
> However, normalization is required to investigate our research hypothesis in either case. If we simply analyze ontologies as they are, and do not show how simple they could have been if expressed differently, then it would be difficult or impossible to show potential expressibility for simple axiom patterns. Axioms may, purely by coincidence, be written in an unnecessarily complex way that could have been expressed with simple patterns. This is one of the major factors that distinguishes our study from a simple compositional analysis.
>
> "The title suggests a qualitative, not quantitative study."\
> The study is somewhat qualitative, as the title suggests! (though certainly not a user survey) By transforming ontologies we investigate the potential for qualitative improvement of development with simple axioms. We think this is a strong point rather than a weak point :)
>
> " I would not introduce n"\
> Thank you for noticing this formatting oversight, we will improve this.

---

### Official Review · AnonReviewer5 · 2021-01-14
**Transformation of OWL axioms in simpler axioms**

**Confidence:** 3
**Impact:** 2
**Design And Technical Quality:** 2

**Review:**


-.-.-
Thank you very much for the response. While some points are clarified main ones are still unclear. Regarding the semantics, I think authors actually admit it is about syntactic analysis "Indeed we perform a semantics-agnostic transformation to syntactically alter the ontology" I still think that the work is about syntactic transformation and the semantics are not analyzed. I tried to get an example of how semantics are preserved asking for a specific example and the response gives logical construct without the mapping to "real world semantics" so it goes again to logical equivalences but no evidence of real-world meaning. As you know I reviewed it before and I think while the experiments are extended, the change of title and hypothesis towards semantics make the content a bit more distant to the expected goal, therefore I would rephrase them removing the semantic part. Also, clarify the part about possible overlaps in ontologies in several batches and the selection of LOV ontologies.

-.-.-


This submission presents an study about OWL transformation patterns over 518 ontologies. The main goal is to proof that most OWL axioms can be replaced by simple patterns. The conversation about the OWL complexity and the real use of its construct is still subject of research and has been reported in the past:

[1] Birte Glimm, Aidan Hogan, Markus Krötzsch, Axel Polleres. OWL: Yet to arrive on the Web of Data? CoRR abs/1202.0984 (2012)

[2] Taowei David Wang, Bijan Parsia, James A. Hendler. A Survey of the Web Ontology Landscape. International Semantic Web Conference 2006: 682-694

[3] A Snapshot of the OWL Web. N Matentzoglu, S Bail, B Parsia - International Semantic Web Conference, 2013 - Springer


The methodology followed is based on the transformation of ontology axioms in simple axioms. This procedure seem to be overcounting more simple axioms after the transformation. Also, it is discussed that the system does not detect disjoint, how does it affect the calculation of "Ontology pattern expressibility"? Is the |O| value lower than the real value?

Regarding the ontology selection there are some questions. It is checked whether all ontologies appear only in one batch? That is, for example, the one anatomy ontology might happen to be included in LOV and Ontobee. Also, there is no explanation about the selection from LOV ontologies, I assume the RDFS ones are discarded but it is not mentioned at all how the number of ontologies is narrowed to 250.


As I actually reviewed this paper previously and would like to mention that authors in this case try to direct the topic towards the "semantic" analysis rather than syntactic. In this version the hypothesis emphasis on the semantic aspect H1: "Almost all axioms in OWL are semantically expressible using the set of simple axiom patterns found in Table 1." However, I'm not convinced that the "semantically" expressivity has been proven, it is rather a syntactic transformation.  In addition, there are some sentences that should be supported:

- "However, there remains an issue that ontologies often vary radically in the way they are syntactically expressed, even if semantically they mean similar things." Could you provide an example?

- "Our motivation for this study is that we believe simple axioms, in general, are easier for non-logicians (for example, domain experts) to understand and utilize for modeling." Has this been tested? In my opinion the transformation are less intuitive than the corresponding OWL axioms in many cases.



It is not clear what "standard way" means in "Thus for EquivalentClasses, DisjointClasses, and DisjointUnion we convert them to sets of SubClass axioms in the standard way."


Regarding the related work, I would suggest to include and compare to results in [1],[2] and [3]. These papers were mentioned in the previous reviews and could be included in the related work also because they are way more related to the topic at hand than for example the systems in the second paragraph of such section (more suitable for the OWLAx paper).



Finally, are classes counted by ontology in table 3? That is, if a URI appears in 2 ontologies in a batch, it is counted twice? I'm asking this because in the previous submission the "Misc" values for  this table are the same as for this one but there is one less ontology (I couldn't find this data in the excel).

**Anonymity:**

Yes, I would like my review to remain anonymous.

**Rating:**

-1: Weak Reject

**Reuse And Availability:**

4: High

**Strong Points:**

- Adequacy for the conference topics
- Software and data provided

**Subreviewer:**

I submitted this review.

**Weak Points:**

- Methodology for transformations and counting
- Novelty: it is true that the approach is different from other experiments but it is another paper saying that most of owl is not used.

---

> ### Author Rebuttal · Authors · 2021-01-29
>
> Thank you for your comments, both this time and in the previous review. Unfortunately our revisions to address the misunderstandings - and also our rebuttal to the previous submission -  do not seem to have resolved your concerns. We will respond to some points you mention.
>
> "The conversation about the OWL complexity..."\
> The nature of axioms as they are in the ontologies is specifically not the point of this study (we mention this several times), though it may be possible to test that in a future evaluation for comparison. We have changed the language in the new version of the paper to make this clear. As a result, while we thank you for the references, of which we are aware, these are entirely different types of evaluations that only superficially seem similar. However since the misunderstanding appears to persist, we will add language to the final version which explains even more explicitly why they are too different for a meaningful comparison.
>
> "This procedure seem to be overcounting..."\
> No overcounting occurs, and we are unable to discern the justification for your claim. The counts before normalization, which we report but do not analyze, are taken directly from the OWLAPI and are shown in Table 3. And the counting after normalization, which has a well documented procedure, and contains, emphatically, no duplicates or overcounts, is correct and displayed in the results. We explain in detail how exactly we count, where is the overcount?
>
> "Is the |O| value lower than the real value?"\
> This could be clarified, thank you for the pointer. There are 2 possibilities, both of which are “real”, and for |O| the size used is implied by the context of the experiment but not explicitly stated. In general the equations for expressibility could apply to a non-normalized ontology so they should not be changed. We study the normalized ontologies since that is a better indicator of pattern use potential. The correct value is used that corresponds to the number of normalized axioms. As we mention in the paper, "the sum of the expressibility sizes of all the axioms in an ontology is always less than or equal to the number of axioms it contains (there are no duplicates)".
>
> "I'm not convinced that the "semantically"..."\
> Indeed we perform a semantics-agnostic transformation to syntactically alter the ontology. This allows us to see which of the axioms, even if they are expressed as complex axioms, might have been expressed with simple axioms. The idea, as we say, is that simple axiom(s) can usually express the same semantics in a way that is syntactically easier for developers and domain experts to understand while meaning the same thing. More generally speaking, in computing the semantics of formal logics is always dealt with using syntactic transformations, like in reasoning algorithms, so we’re also a bit puzzled by the objection. Perhaps the issue is a confusion between “expressivity”, which we do not study, and “expressibility”, which we do study?
>
> "It is not clear what "standard way" means…"\
> By "standard way", we mean the standard functions in the OWLAPI, which themselves are documented and also appear in the definitions in the OWL 2 structural specification. We will be more explicit about this.
>
> "Finally, are classes counted ..."\
> 	One of the ontologies in Misc for the previous paper threw errors and did not load, so it was removed for this evaluation. The group counts are shown as summed totals (it would be impractical to show every ontology count in the paper, though that is in the spreadsheet).
>
> "Could you provide an example?"\
> An ontology made by a novice with simple patterns from modeling software might be different from the same ontology written by a logician who needs no tools and writes complex expressions naturally. Additionally, because the normalization in this study is syntactic, every ontology we use before and after the change is an (artificial) example. To be even more basic, think of $C \sqcup D \sqsubseteq \bot$ versus $C \sqsubseteq \lnot D$ versus $D \sqsubseteq \lnot C$ etc.

---

### Official Review · AnonReviewer4 · 2021-01-18
**new metrics about ontology axiomatization**

**Rating:** 1
**Confidence:** 3
**Impact:** 3
**Design And Technical Quality:** 4

**Review:**

This paper presents two metrics to evaluate the complexity of ontology.

The description logics identified is SROIQ(D).

The methods is based on a definition of simple axiom patterns.
The authors want to test the hypothesis that ontology used simple patterns.

Simple pattern should have at most 3 elements.
A list of 17 simple patterns is presented : subclass, existential, domain, range,... structural tautology.
Note that cardinality restrictions are not part of simple patterns. Only Functional restriction is part of simple pattern. I would appreciate an explanation why cardinality above one is not considered.

The proposed metric untittled « ontology pattern expressivity » (oep) counts the frequency of those patterns in one ontology. This value is normalized by the number of axioms found in the ontology.

The metric is extended to a set of ontologies with two others metrics :
the average ontology pattern expressivity (the average of oep for the set of ontologies)  and the average axiom pattern expressivity (the oep evaluated on the merging ontology which is the sum of axioms of all ontologies from the set).

To compute those metrics first the ontology’s axioms are normalized. One of the normalization process is a decomposition of conjunction and disjonction into subclass axioms. Thus a complex axiom (C subClassOf D and E) is divided into simple axioms (C subClassOf D ; C subclassOf E). What is the impact of this decomposition  on the subclass pattern ?

A set of 518 ontologies has been studied :
250 onto from LOV, 80 ODP, 4 Hydrography, 2 Anatomy, 7 Conferences,171 medical (ontobee) and 4 miscelenious.

table 3 presents the average statistics on these ontologies before normalization. As far as I understand it is not the average number of classes per ontology but the total number of classes fro the whole set of ontologies. I found the number very hight…
@Authors : could you check the table 3 and be more clear about the value of each line.

I would appreciate to see the same table after the normalization process. In order to see the impact of the normalization and the decomposition of complex axioms.

Based on the metrics the authors can argue that simple axioms are the most used axioms in the ontology set.

The figure 1 and 2 present in details by pattern their frequencies in the whole set of ontologies. Thus the most used pattern are subclassOf, and existential.
I found these figures interesting I was expecting that domain and range are more frequent.



Then the study classified the ontology by profile (EL, QL, RL and DL).
The two metrics are computed for the 4 set of ontologies.
I have to say that I found more interesting the study to identify which pattern is most used than the study about the proposed metrics.

The proposed metrics can be adapted to be used in ontology repository when we look for an ontology. User  want to retrieve an ontology with some specific pattern (subClass or Domain or Range) in order to reused this ontology for data interoperability purpose or find a more axiomatized ontology to do reasoning.

In the discussion the authors should identified the usage of ontologies. For example LOD ontology (like in LOV) are used to improve the data interoperability. Thus ontology is just a definition of class and property name with few axioms. If an ontology want to be reused easily it should not provide any constraints and just a vocabulary. This ontology will used maybe just subClassOf Pattern, domain and range. Note that DublinCore on of the most used vocabulary has no domain and range axioms. The study about ontology could compare the frequency of reused (from LOV repository incoming links)  from the type of patterns found.

If the ontology is used for reasoning purpose or scientific theory modeling (like the work on BFO that is used in biology science to define a set of biology description theories) then the ontology will contains complex class axioms. See the crop ontology repository that is based on BFO. This is this type of ontology that are difficult to build and should explain and clarify  their design pattern.

-------------------------------------------------------------------

Thanks the authors for their explanation. I still find this paper interesting for eswc audience. I like very much the figure 1 and 2 about the frequency of predefined axioms patterns. For me it is the strong point of this paper.
I do not know what to to with the proposed metrics, I would adapt them by selecting some specific axiom pattern , like subClassOf or domain and range restriction and then compute their frequencies in a set of ontologies... etc...
I agree with other reviewers that the concept of simple axiom pattern should be better defined.
I hope that the authors would evaluate the impact of the normalization process in their metric computation.

**Anonymity:**

Yes, I would like my review to remain anonymous.

**Reuse And Availability:**

3: Medium

**Strong Points:**

the paper is clear

a set of 17 simple axiom pattern are proposed

3 new metrics have been proposed to evaluate the axiomatization of ontology and ontologies set.

those metrics were used to compare 518 ontologies available on the LOD

the code to compute the metrics is available on a github

the hypothesis that most part of ontologies reused simple pattern have been evaluated. Most part of ontology reused subClassOf and existential axioms.

I like the figure where the most used pattern is identified. This is for me the strong point of this study.
I do not find usefull those metrics in a general study! But those metrics can be adapted and reused  in any ontology repository to evaluate the type of ontology we could find: skos, simple axiom pattern or complex axiom pattern.

**Subreviewer:**

I submitted this review.

**Weak Points:**



I do not find the list of ontologies used in the study on the github.
the normlization process seems to have an impact on the subClassOf pattern which is not evaluated.
All disjonction axioms are not identified by this method, the authors provide some explanation about this topic.
only functional axioms are evaluated not all cardinality restriction axioms.

axiom pattern is far away from design pattern and ontology engineering is more about how to reuse design pattern correctly (identify them and reused them). I do not see how simple pattern can be used in ontology engineering method.

---

> ### Author Rebuttal · Authors · 2021-01-29
>
> Thank you for the comments. There are a few points you mention that we could have explained better. We will respond to those in order that they appeared.
>
> "I would appreciate an explanation why cardinality above one is not considered."\
> 	Rather than devise a new set of patterns arbitrarily, we evaluate a previously published set of patterns.The patterns we use are taken from the ontology tool OWLAx. We could certainly add a cardinality pattern in a future study!
>
> "What is the impact of this decomposition  on the subclass pattern ?"\
> "@Authors : could you check the table 3..."\
> "the normlization process seems to have an impact ..."\
> 	It is a good idea to also show the counts after normalization, this will be added to our tables. However, this study should be distinguished from a standard compositional analysis of ontologies. For this reason we focus not on how ontologies are expressed with simple axioms, but rather how they could have been expressed with simple axioms. This is why we say "expressible" and "can be expressed", avoiding "are expressed".\
> The counts in Table 3 are correct. They were obtained directly from the OWLAPI before normalization. Many medical ontologies are very very big containing thousands of classes, GO especially. Normalization would only impact the axiom counts, which we will add to the table.
>
> "The study about ontology could compare the frequency of reused (from LOV repository incoming links) from the type of patterns found."\
> 	This is an interesting idea, but would require the design of a follow-up study. The present study uses single OWL ontologies, specifically exploring the potential usefulness of developing OWL ontologies with simple patterns. DublinCore, for instance, does not obviously apply because it is purely RDF. Even using LOV requires extracting OWL ontologies with a script that follows ontology links from the LOV dump, many of which do not work correctly because a significant number of links in the LOV dump have inconsistent destinations and/or don’t resolve.
>
> "I do not find the list of ontologies used in the study on the github."\
> 	It is a good idea to list the ontology names, we will put this in the github readme. There is a folder called "OWL" in the GitHub that contains every file used in the analysis, with the exception of GO which is too large to fit.
>
> "If the ontology is used for reasoning … then the ontology will contains complex class axioms."\
> 	We proposed, and we believe also show, that axioms could usually be expressed with simple patterns that are easy to understand and use. Obviously this isn’t always the case, as you say, and sometimes complex axioms are necessary. However, even GO, which is an extremely large and very complex ontology, has more than 60% expressibility with simple axiom patterns using our basic analysis - this is one of the results that also surprised us. If over half of the hardest ontology axiomatization challenges could be simplified this would be beneficial to ontology engineering.
>
> "I do not see how simple pattern can be used in ontology engineering method."\
> 	We distinguish axiom patterns (as in this paper) from ontology design patterns. In the end, every ontology (whether designed from ontology design patterns or by some other method), will require concrete axioms, and our study shows that most axioms can be expressed with simple axiom patterns. We will make this connection more clear in the final version of the narrative. For details on how the OWLAx tool (and thus our axiom patterns) fit into an ontology design pattern based ontology modeling approach - see http://www.semantic-web-journal.net/content/modular-ontology-modeling .

---

### Decision · Program_Chairs · 2021-02-23

**Decision:**

Accept with shepherding

**Comment:**

The reviewers agreed that the paper  presents an interesting study on prevalence of simple axioms in existing ontologies.

The reviewers highlighted several issues that must be addressed by the authors in the camera ready of the paper.
More precisely, we recommend the authors to:

1.  Revise the  presentation to better explain terminology.
2.  Define the key terms to facilitate the interpretation of the key contribution.

3.  Discuss of the limitations of the evaluation.

4.  Make a clearer distinction between axiom patterns (as in this paper) from ontology design patterns.

5. Emphasis why the references cited by the reviewers, of which the authors are aware,  present entirely different types of evaluations that only superficially seem similar.

6. What does the value of  |O |refer to ?

7. deep reading of the paper to check all editing tips.

Although the reviewers point to different clarifications, they believe the necessary changes can be feasibly made by the authors. The authors have to include all clarifications in the final version of the paper.